# The Growth of Graphene on Ni–Cu Alloy Thin Films at a Low Temperature and Its Carbon Diffusion Mechanism

**DOI:** 10.3390/nano9111633

**Published:** 2019-11-17

**Authors:** Yibo Dong, Sheng Guo, Huahai Mao, Chen Xu, Yiyang Xie, Chuantong Cheng, Xurui Mao, Jun Deng, Guanzhong Pan, Jie Sun

**Affiliations:** 1Key Laboratory of Optoelectronics Technology, College of Microelectronics, Beijing University of Technology, Beijing 100124, China; donyibo@emails.bjut.edu.cn (Y.D.); xieyiyang@bjut.edu.cn (Y.X.); dengsu@bjut.edu.cn (J.D.); guanzhongpan@emails.bjut.edu.cn (G.P.); 2Department of Industrial and Materials Science, Chalmers University of Technology, 41296 Gothenburg, Sweden; sheng.guo@chalmers.se; 3Materials Science and Engineering, KTH Royal Institute of Technology, Brinellvägen 23, 10044 Stockholm, Sweden; huahai@kth.se; 4Thermo-Calc Software AB, Råsundavägen 18, 16967 Solna, Sweden; 5State Key Laboratory of Integrated Optoelectronics, Institute of Semiconductor, Chinese Academy of Sciences, Beijing 100083, China; chengchuantong@semi.ac.cn (C.C.); maoxurui@semi.ac.cn (X.M.); 6Quantum Device Physics Laboratory, Department of Microtechnology and Nanoscience, Chalmers University of Technology, 41296 Gothenburg, Sweden

**Keywords:** transfer-free, lithography-free, graphene, chemical vapor deposition, insulating substrate, low temperature growth

## Abstract

Carbon solid solubility in metals is an important factor affecting uniform graphene growth by chemical vapor deposition (CVD) at high temperatures. At low temperatures, however, it was found that the carbon diffusion rate (CDR) on the metal catalyst surface has a greater impact on the number and uniformity of graphene layers compared with that of the carbon solid solubility. The CDR decreases rapidly with decreasing temperatures, resulting in inhomogeneous and multilayer graphene. In the present work, a Ni–Cu alloy sacrificial layer was used as the catalyst based on the following properties. Cu was selected to increase the CDR, while Ni was used to provide high catalytic activity. By plasma-enhanced CVD, graphene was grown on the surface of Ni–Cu alloy under low pressure using methane as the carbon source. The optimal composition of the Ni–Cu alloy, 1:2, was selected through experiments. In addition, the plasma power was optimized to improve the graphene quality. On the basis of the parameter optimization, together with our previously-reported, in-situ, sacrificial metal-layer etching technique, relatively homogeneous wafer-size patterned graphene was obtained directly on a 2-inch SiO_2_/Si substrate at a low temperature (~600 °C).

## 1. Introduction

Graphene, since its “discovery” in 2004, has attracted worldwide attention because of its excellent properties, such as high carrier mobility, high mechanical strength and high transmittance [1,2]. Several methods have been developed for graphene synthesis, including mechanical exfoliation [3], liquid phase processes [4,5,6], epitaxy on SiC [7,8] and chemical vapor deposition (CVD) [9,10,11,12]. Among them, CVD is an efficient method for preparing high-quality, large-area graphene films, which makes it one of the most commonly used methods. The CVD method generally uses a catalytic metal substrate like Cu, Ni, etc. [12]. Ni, for example, is a commonly used catalytic metal, but the biggest problem for Ni as catalyst is its high carbon solid solubility. More carbon atoms than necessary are dissolved in Ni when graphene grows, and they will segregate when the temperature is cooled down, resulting in the formation of multilayer graphene [13]. Carbon solid solubility is a temperature-dependent quality. As the growth temperature decreases, the carbon solid solubility in Ni decreases. If we lower the growth temperature while making sure the necessary amount of carbon is dissolved, can we obtain uniform monolayer graphene on the nickel surface by suppressing the carbon segregation? That line of thinking motived the current work.

In this work, we studied the growth of graphene by plasma enhanced CVD at a low temperature (~600 °C). Thin films of Ni–Cu alloy were chosen as catalysts. Cu was used to increase the carbon diffusion rate (CDR), while Ni was used to provide high catalytic activity. The plasma power was optimized, since it was found to have a significant impact on the graphene quality. We found that at a low temperature, the CDR, instead of the carbon solubility, played the dominant role in determining the number of graphene layers and uniformity. The CDR sharply decreased at decreasing temperature, leading to inhomogeneous and multilayer graphene formation. Nevertheless, relatively homogeneous, wafer-sized patterned graphene was still achieved after process parameter optimization. Our findings will be useful for further growth optimization, especially at low temperatures, because the information available in literature that can serve as guidelines for low temperature graphene growth is very limited.

## 2. Materials and Methods

### 2.1. Graphene Synthesis

The substrate we used was heavily doped Si with 300 nm SiO_2_. The equipment used for the deposition of Ni and Cu films was a Denton Vacuum sputtering platform. Radio frequency source was used for the sputtering. Ni–Cu alloy was prepared by depositing Ni and Cu films successively, followed by annealing at 600 °C for 40 min in a high-vacuum annealing furnace to make them alloy and grow the grains. When sputtering, the substrate was kept at 30 °C. The sputtering power of Ni and Cu is 300 W, the corresponding direct-current biases are 142 V for Ni and 188 V for Cu; the sputtering rate of Ni is about 1.6 Å/s; and the rate of Cu is about 3 Å/s. The total metal thickness was 300 nm. The compositions of Ni–Cu alloy are controlled by adjusting the thickness of Ni and Cu films. For example, the ratio of Ni to Cu 1:10 can be obtained by setting the thickness of Ni film to be 30 nm and the thickness of Cu film to be 270 nm.

The growth equipment was a vertical, cold wall CVD furnace (Black Magic, Aixtron, Herzogenrath, Germany). First, the temperature rose to 600 °C at a rate of 200 °C/min in H_2_ atmosphere. The pressure was controlled to be 15 mbar. Then, 960 sccm Ar, 20 sccm H_2_ and 5 sccm CH_4_ were introduced into the chamber and the plasma (50 W, 100 kHz) was ignited at the same time. The plasma was maintained for 5 min to grow graphene. Finally, the furnace was cooled down at 200 °C/min. The whole growth process (including heating and cooling) took approximately 15 min. The schematic illustrations of the growth equipment and growth process are shown in Appendix A.

We measured the roughness of the Cu–Ni films by a stylus profiler (Bruker Inc., Tucson, AZ, USA). Before annealing, the average roughness and the root-mean-square roughness of the films were Ra = 3.03 nm and Rq = 3.76 nm, respectively. After annealing and graphene growth, the average roughness and the root-mean-square roughness of the films were Ra = 4.41 nm and Rq = 5.62 nm, respectively. On the whole, the surface roughness of the Ni–Cu alloy film did not change much, due to the relatively low annealing and growth temperature.

### 2.2. Metal Sacrificial Layer Etching

First, the PMMA (poly(methyl methacrylate)) supporting layer was spun on the substrate (3000 r/min, 30 s); that was followed by baking the substrate at 150 °C for 5 min. Then, the sample was immersed in a Ni–Cu alloy etchant (CuSO_4_:HCl:H_2_O = 10 g:50 mL:50 mL) for about 35 min. The thickness of PMMA was about 100 nm. After the etching of the metal, the sample was immersed in deionized water for 15 min and blown dry. Then, the sample was baked at 150 °C in air for 10 min to improve the graphene-substrate adhesion. In the end, the PMMA was removed by acetone.

## 3. Result and Discussion

### 3.1. Diffusion Rate of Carbon Atoms on Metal Surface

Ni is known to be more catalytic than Cu in graphene growth [14]. Previously, we have studied the growth of graphene at 800 °C and found that with the increase of Ni composition in Ni–Cu alloys, the quality of graphene increased gradually [15]. However, the carbon solid solubility of Ni–Cu alloy also increases with the increase of Ni composition [16]. We found that when the percentage of Ni reached 50%, graphene showed notable multi-layer areas [15]. Therefore, at 800 °C, we controlled the proportion of Ni to Cu to be 1:2.

Carbon solid solubility is positively correlated with temperature. At a 600 °C growth temperature, therefore, the carbon solid solubility was to presumably be lower than that at 800 °C. Our initial proposition was, thus, to increase the Ni composition in Ni–Cu alloy at 600 °C, which would improve the quality of graphene, without significantly damaging the uniformity of graphene.

However, we found that the experimental results were not exactly in line with our expectations. Figure 1 shows the Raman spectra and corresponding optical images of the graphene grown on SiO_2_/Si substrates catalyzed by Ni–Cu with different compositions. Plasma was applied to assist the carbon source pyrolysis. At a growth temperature of 600 °C, we did observe that the quality of graphene increased with the increase of Ni composition, which was mainly manifested by the decrease of D peak in Raman spectra of the graphene. The D peak represents inter-valley, defect-induced resonant scattering [17]. However, the number of graphene layers still increases significantly with the increase of Ni composition. Graphene with different numbers of layers shows different contrasts on 300 nm SiO_2_ [18], so the number of layers can be roughly recognized by observing the color contrast by optical microscopy. We observed that when the Ni–Cu ratios were larger than 1:2 (Figure 1d,e), the color of graphene appeared to be dark and it was inhomogeneous. In particular, when the ratio reached 2:1 (200 nm Ni + 100 nm Cu), the color of the graphene (see Figure 1e) was quite dark, indicating that the thickness of graphene was rather great. It shows that with the increase of Ni percentage, the graphene grown at 600 °C becomes thicker and appears with uneven layer distribution. Even though such a phenomenon was observed in our previous work [15] on the graphene growth at a higher temperature, i.e., 800 °C, it seemed contrary to our expectations, considering the limited carbon solid solubility at low temperatures. We found that this phenomenon could no longer be explained by the known mechanism of carbon atom segregation. According to Laurent Baraton et al. [19], at 725 °C, the number of carbon atoms dissolved in 200-nm-thick Ni is about 8 × 10^15^/cm^2^, which corresponds to the number of carbon atoms required to form bilayer graphene. At 600 °C, as in our condition, the number of carbon atoms that can be segregated from the Ni2Cu1 (representing a Ni to Cu ratio of 2:1) alloy should be much less than that of bilayer graphene. However, atomic force microscopy (AFM) measurements show that the maximum thickness of the graphene grown on the alloy was about 88 nm (Figure 2b). This thickness is equivalent to hundreds of layers of graphene (0.34 nm for a monolayer [20]), which is much larger than the number of carbon atoms that can be dissolved in Ni–Cu alloy and is even thicker than graphene catalyzed by pure Ni at high temperatures. By contrast, the graphene catalyzed by a Ni1Cu3 (representing a Ni to Cu ratio of 1:3) alloy was only about 1.9 nm thick (Figure 2a) with an even distribution (Figure 1b) under the same growth conditions.

To further understand the growth mechanism at a low temperature, we performed more experiments. We think that the low CDR on the surface of Ni relative to that on Cu plays an important role. We shortened the growth time to 30 s and observed the graphene films grown on Ni1Cu3 (low Ni composition) and Ni2Cu1 (high Ni composition) alloys. As shown in Figure 2c, the graphene on the Ni1Cu3 alloy grew into a continuous film in the short time and the graphene was uniform. In contrast, the graphene grown on the Ni2Cu1 sample was still discontinuous (Figure 2d). Meanwhile, the graphene was already very thick. It was found by electrical measurements that the graphene grown on the Ni1Cu3 sample was conductive, while the graphene grown on the Ni2Cu1 sample was not conductive, which further verified that graphene in Figure 1b was continuous, while the graphene in Figure 1e was discontinuous. We also observed the samples by scanning electron microscope (SEM). As shown in Figure 2e,f, the graphene in Figure 2e is a continuous film, while the graphene in the Figure 2f is flaky. There is no connection among the flakes, indicating it is discontinuous. The growth rate of graphene on a metal surface is mainly related to the growth conditions, the metal catalyst, nucleation point density and CDR on the metal surface. All the samples in our experiment were grown in the same batch, excluding the influence of different growth conditions. We observed that the average grain size of Ni film is smaller than that of Cu at a same thickness (Appendix A). In addition, the catalytic activity of Ni is higher than that of Cu [14]. These observations indicate that, on the surface of the Ni2Cu1 alloy, the graphene nucleation point density should be higher and the cracking rate of methane should be faster. However, within the same growth time, the graphene on the surface of Ni1Cu3 alloy reached full coverage, while the graphene on the surface of Ni2Cu1 did not. This phenomenon indicates that carbon atoms diffuse rapidly on the surface of the low Ni composition alloy (e.g., Ni1Cu3), so they can grow into a continuous graphene film in a very short time. When the graphene fully covers the metal surface, which isolates the contact between methane and the metal, the catalytic action of the metal disappears or at least drastically decreases, and the graphene growth rate is greatly slowed down. For alloys with a high Ni composition, carbon atoms diffuse slowly on their surfaces, and therefore, they accumulate near the graphene nucleation center. Therefore, it takes a longer time to form a continuous graphene film. The catalytic action of the metal will continue until the metal is fully covered, resulting in very thick and also uneven graphene. Analogically, the scenario can be understood as we pour a glass of water on a smooth plane, and the water flows quickly on the plane to form a uniform water film. However, if we pour a glass of sand on the same plane, the sand finds it more difficult to form a uniform layer. As shown in Figure 2g,h, the diffusion of carbon atoms on Cu surface is equivalent to “pouring water on a smooth plane”, while the diffusion of carbon on Ni is equivalent to “pouring sand”.

In order to verify our assumptions, we tried to estimate the relative surface CDR theoretically. In similar external environments, the relative CDR on the surface of Ni–Cu alloys is mainly determined by the alloy composition. There may be different mechanisms and factors for the diffusions inside a bulk and along a surface. The absolute quantity differs a lot between the bulk/surface-diffusivity even for the same material. Nevertheless, the surface CDR is in principle correlated to the bulk CDR. Hancke and Haugsrud [21] confirmed the positive correlation between surface kinetic and bulk diffusion. Therefore, we may assume the composition-dependence of surface CDR is positively correlated to the bulk CDR. In the present case, the bulk CDRs we calculated for various Ni–Cu alloy compositions at 600 °C may have served as a semi-quantitative factor. The calculation was made based on the Thermo-Calc Software ^®^, <www.thermocalc.se>, version 2019a, using the MOBCU2 atomic mobility database. As shown in Figure 3a, with the increase of Ni composition, the CDR in the alloy decreases rapidly. The CDRs in pure Cu and Ni are 2.5 × 10^−12^ m^2^/s and 3 × 10^−14^ m^2^/s respectively, which are about two orders of magnitude different. In Figure 1, we observe that when the ratio of Ni to Cu reaches 1:1, the graphene layers are obviously thick and begin to become inhomogeneous. Thus, from Figure 3a we can roughly get that a critical diffusion rate of 2.5 × 10^−13^ m^2^/s is needed to ensure the uniformity of graphene layers. We also calculated the bulk CDR at 800 °C (Figure 3b). The temperature and diffusion rate are positively correlated. It can be seen that when the temperature increases by 200 °C, the CDR in pure Ni has already exceeded the critical value of 2.5 × 10^−13^ m^2^/s, so the CDR will not affect the uniformity of graphene at 800 °C. In this sense, it is understandable that little attention has been paid to CDR in previous reports where the growth temperatures are high. To our knowledge, this is the first work to study the effect of CDR on the thickness and uniformity of graphene in low temperature growth.

The high CDR in/on Cu relative to that of Ni is attributed to the following factors. The diffusion coefficient is given as D = D_0_exp (-Q/k_B_T), in which D_0_ (the frequency factor) is approximately temperature-independent. Q is the diffusion activation energy, k_B_ is the Boltzmann constant and T is the absolute temperature in Kelvin. We see that D_0_ depends on the crystal lattice. In the present case, Cu has a larger D_0_ value, mainly due to a larger lattice constant than Ni. The activation energy Q sitting in the exponent function should have a stronger impact on D, in comparison with D_0_ which is proportional to D. The high diffusivity in Cu can, therefore, be attributed to two factors: (1) a larger lattice constant, and (2) a lower diffusion barrier which can be further interpreted based on the filling status of the d electron shell. Hu et al. [22] studied the solubility and mobility of C interstitials in transition metals used for the growth of 2D materials. They found the interstitial formation energies of C in Cu are much higher than those in Ni, which indicates Cu is less likely to accommodate C. Consistently, they confirmed that the migration barriers of C interstitials in Cu are lower than those in Ni, indicating an easier mobility for C in Cu.

Thus, the performances of Ni–Cu alloys at low-temperature growth are dramatically different from the previous reports at high temperatures [23,24,25,26]. Ni is mainly used to provide high catalytic activity to improve the quality of graphene grown at low temperatures, while Cu is mainly used to improve the CDR on the surface of the alloy to ensure a thin and uniform graphene film. By compromising the quality and uniformity of graphene, we found that the Ni–Cu ratio of 1:2 was the best ratio. This ratio guarantees both the uniformity and relatively high quality of graphene.

This work also gave us inspiration for our follow-up work. We can further explore alloys with higher catalytic activity and higher CDR on its surface simultaneously, in order to obtain even more uniform and higher quality graphene at low temperatures. The selection of new alloys must meet the following requirements: (1) high catalytic activity, (2) fast CDR in the metal and (3) forming a homogenous single-phase solid solution alloy. We screened CDR in various metals [22] and the catalytic activities of these metals [14], and inspected the phase diagrams of various binary alloys. Finally, three potential alloy catalysts for low temperature growth were identified; namely, Cu–Pd, Au–Ni and Co–Ru. The detailed analysis process is shown in the Appendix A. In the future, we will try to use these alloys for graphene growth.

### 3.2. Plasma Enhancement

We also studied the influence of plasma on the quality of graphene. The purpose of applying plasma during the growth is mainly to accelerate the low temperature cracking of methane. Interestingly, we found that plasma is also very important to improving the quality of graphene. In fact, graphene can also be grown on the surface of Cu–Ni alloys at 600 °C even without plasma. It was found that the lowest growth temperature catalyzed by pure Ni without plasma is around 450 °C (Appendix A). However, without plasma participation, the quality of graphene decreases significantly (Figure 4a). This is mainly because without plasma, it is difficult for methane to achieve complete pyrolysis. Therefore, during growth, the starting time of plasma is best synchronized with the time of methane inflow. If we turn on the plasma after methane has been introduced for some time, the graphene has already grown on the metal surface, resulting in poor quality of the graphene. The plasma power also affects the quality of graphene. As shown in Figure 4b, we experimented with four levels of power—25 W, 50 W, 100 W and 150 W, with a frequency of 100 kHz. Raman results show that 50 W is the best because the D peak of the graphene is the smallest. For higher power, the growth reaction will become very violent, leading to the increase of defects in graphene. The plasma electrodes we used are located below the heater (Figure 4d). Therefore, for lower power, it is difficult for the plasma electric field to cover the surface of the heater, resulting in incomplete pyrolysis of methane and a decrease of the graphene’s quality. The distance between the plasma electrode and the heater will also directly affect the electric field strength on the heater’s surface. When the plasma electrode is closer to the heater, the optimal power should be reduced appropriately. The thickness of graphene is also affected by the plasma power. It was found that with a higher plasma power, the graphene was thicker.

The growth equipment we used included a vertical, cold wall graphene growth system (Black magic, Aixtron). As shown in Figure 4c, the plasma electrode was made of Cu upon our reconstruction and the heater was made of graphite, which was heated by Joule heat. When growing, a heavily doped silicon wafer was placed on the heater and the sample was placed on the silicon. Figure 4d was taken during the growth process. The methane/hydrogen ratio was 5 sccm/20 sccm. Methane was used to grow graphene, and hydrogen was used to etch the defects in graphene to improve its quality [27,28]. These two effects are both dynamic processes, and the reaction speeds of the two were greatly accelerated with the enhancement of plasma. The methane/hydrogen ratio of 5 sccm/20 sccm was chosen to ensure the stability of the chamber environment after multiple growths (Appendix A).

### 3.3. Wafer-Scale Patterned Graphene Direct Growth

One of the applications of low temperature growth is the combination with direct growth technology, which makes it possible to directly grow graphene on a variety of substrates that cannot withstand very high temperatures. We combined this 600 °C graphene growth method with our previously-reported in situ growth technique [15,29] and obtained wafer-scale patterned graphene directly on the SiO_2_/Si substrate.

First, we need to give a brief introduction of our in situ growth technology. This method uses a sacrificial metal layer to assist growth. The essence of the growth method is the etching mechanism of the metal sacrificial layer. As shown in Figure 5a, we deposit patterned metal film on insulating substrates, such as quartz, SiO_2_/Si and sapphire. The graphene is grown on the metal surface. After that, a PMMA supporting layer is spun on the sample. Therefore, the metal sacrificial layer is completely covered by the PMMA/graphene films. Interestingly, we find that when the sample is immersed in the metal etchant, the etchant can efficiently penetrate the PMMA and graphene to accomplish the metal etching (Figure 5b). As shown in Figure 5c,d, after the etching of the metal, the graphene film will fall on the substrate because it is firmly fixed on the substrate surface by PMMA. Then, the sample is immersed in deionized water for cleaning. After the sample is dried, the PMMA/graphene films are baked at 150 °C for 15 min to enhance the adhesion to the substrate. Finally, the PMMA is removed by acetone. In that manner, in-situ, transfer-free growth of graphene is realized. We call this method “in-situ” and “transfer-free” because the graphene is grown and finally used on the same one substrate and there is not any transfer during the whole process. Besides, the graphene patterns and the positions on the substrate are identical to those of the sacrificial metal layer. By this method, we can grow graphene films of any pattern at any position on the substrate [15,29].

Using this method, a patterned graphene film was grown directly on a 2-inch SiO_2_/Si substrate. The plasma power was selected to be 50 W; the ratio of methane to hydrogen was 5 sccm/20 sccm; and the growth time was 5 min. Figure 5e is a photograph of the sample after the spinning of PMMA and before etching of the alloy. Figure 5f is a photograph of the final, wafer-scale patterned graphene sample. Figure 5g is an optical image showing the graphene in a local area of the sample. The graphene we grew exhibited a relatively uniform color, indicating that the graphene was macroscopically uniform. The graphene had patterns. Limited by the size of the heater (Figure 5c), 2 inches is the largest area we can grow. In principle, our method can be scaled up. Figure 5h,i show the Raman mapping data measured in a 50 × 50 µm area (12 × 12 points). Figure 5h is the mapping of D/G peak ratio, where I_D_/I_G_ lies in the range of 0.12–0.69. Figure 5i is the mapping of G/2D peak ratio with I_G_/I_2D_ of typically 0.97–3.6.

## 4. Conclusions

In this work, the effect of CDR on the thickness and uniformity of graphene in low temperature growth has been revealed for the first time. The uniformity and quality of graphene can be improved by proper selection of the alloy compositions. Ni–Cu alloy was used in our work, where Cu acted to increase the CDR, while Ni provided high catalytic activity. Meanwhile, we found that the plasma power can have great influence on the quality of graphene. Finally, based on optimizing the growth parameters and combining them with the in-situ growth method, we obtained a wafer-scale graphene film directly on SiO_2_/Si substrate at 600 °C with relatively good quality and uniformity. Low temperature growth can expand the range of substrate choices and reduce the effect of temperature on substrates, which is a future direction for the development of direct graphene growth. We expect the work presented here will promote the application of directly grown graphene at low temperatures.

## Figures and Tables

**Figure 1 nanomaterials-09-01633-f001:**
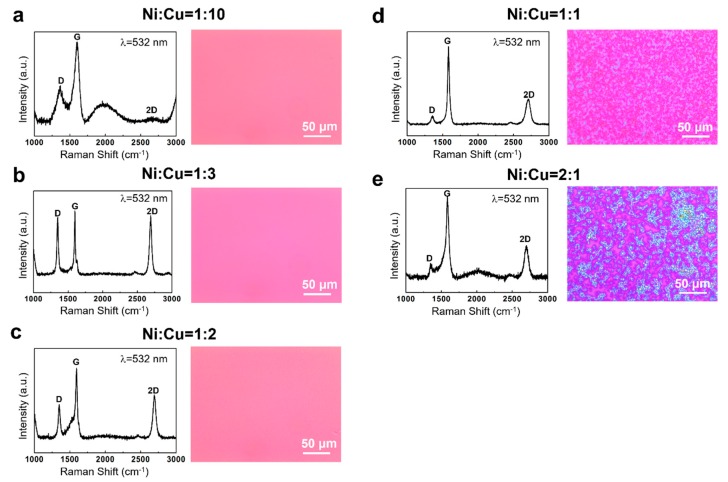
(**a**)–(**e**) The Raman spectra and corresponding optical images of the graphene on SiO_2_/Si substrates catalyzed by Ni–Cu alloys with different compositions (Ni:Cu = 1:10 (**a**), 1:3 (**b**), 1:2 (**c**), 1:1 (**d**) and 2:1 (**e**), respectively).

**Figure 2 nanomaterials-09-01633-f002:**
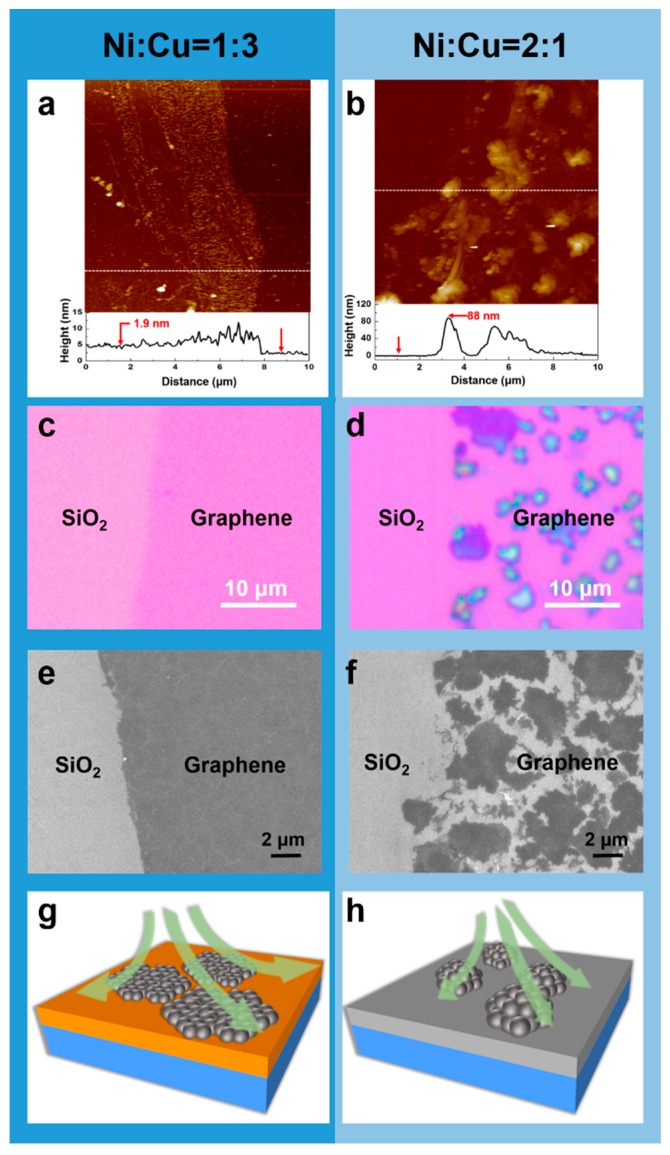
(**a**,**c**,**e**) The AFM, optical and SEM images of the graphene grown on the Ni1Cu3 alloy, respectively. (**b**,**d**,**f**) The AFM, optical and SEM images of the graphene grown on the Ni2Cu1 alloy, respectively. (**g**,**h**) The diffusion mechanism of carbon atoms on Cu and Ni surfaces, respectively.

**Figure 3 nanomaterials-09-01633-f003:**
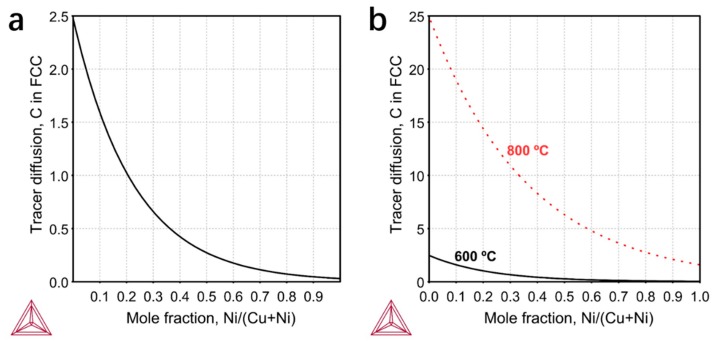
Theoretically determined tracer diffusivity (10^-12^ m^2^/s) for (**a**) the bulk diffusion rate of carbon atoms in alloys with different Ni–Cu ratios at 600 °C. (**b**) The bulk diffusion rate of carbon atoms in alloys with different Ni–Cu ratios at 600 °C and 800 °C.

**Figure 4 nanomaterials-09-01633-f004:**
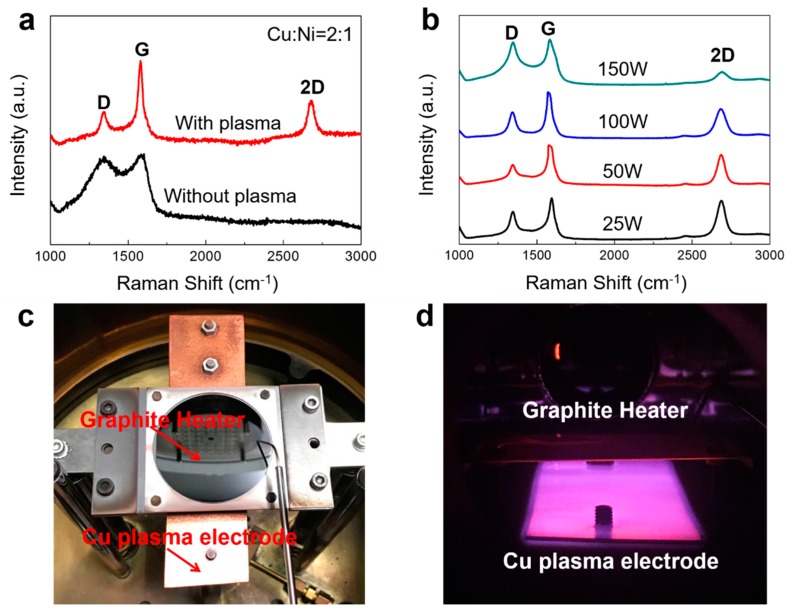
(**a**) Raman spectra of the graphene grown with and without plasma at 600 °C. (**b**) Raman spectra of the graphene grown with different plasma powers. (**c**) The photograph of the growth chamber. (**d**) The photograph taken during the growth process.

**Figure 5 nanomaterials-09-01633-f005:**
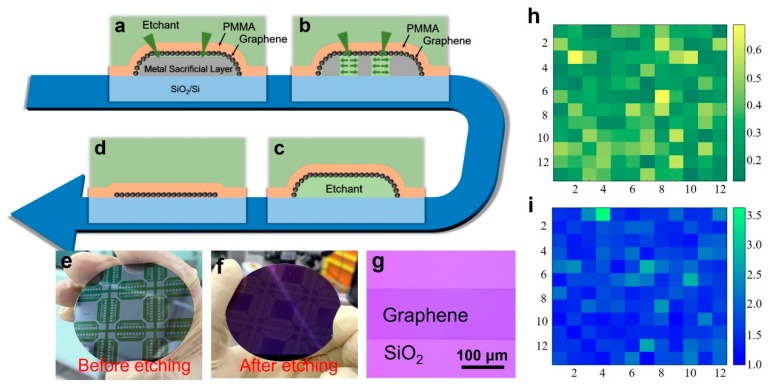
(**a**–**d**) Schematic illustration of the etching process of our in-situ, transfer-free growth method. (**a**) After the graphene growth, PMMA coating is spun on the sample surface. (**b**) When the sample is immersed in the metal etchant, the etchant can efficiently penetrate through the molecular gap of PMMA and the grain boundary of the graphene to achieve the metal etching. (**c**,**d**) After the metal is completely etched away, the PMMA/graphene films will fall on the substrate. (**e**,**f**) Wafer-level graphene growth. The photograph of the sample before (**e**) and after (**f**) the metal sacrificial layer etching. (**g**) An optical image of the graphene at 1000× magnification. (**h**,**i**) Raman mapping of the D/G and G/2D ratios of the graphene over 50 × 50 μm.

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
