# Peer review of "The Growth of Graphene on Ni–Cu Alloy Thin Films at a Low Temperature and Its Carbon Diffusion Mechanism"

_nanomaterials, 2019, doi:10.3390/nano9111633_

Round 1
Reviewer 1 Report
The work presented shows interesting results on the formation of graphene on Ni-Cu alloy thin films. A few comments are shown below:
The title is somewhat limited in scope, since it is not just carbon diffusion rate that is studied; a better title might be "The growth of graphene on Cu-Ni alloy thin films". There should be more information on the deposition parameters for the Cu-Ni films, such as substrate temperature, bias, deposition rate and film thickness. The surface roughness of the Cu-Ni films could be an important parameter - has this been measured? The use of bulk diffusion rates for carbon in Cu-Ni alloys to make conclusions about surface diffusion rates has limitations. While this is noted in the paper, it is the central hypothesis of the work and therefore justified more completely in terms of surface vs. bulk diffusion mechanisms.Author Response
Please see the attachment.

Reviewer 2 Report
Many interesting results are provided in this article. The low temperature growth of graphene is very important matter. The results can be expected to contribute to production of new electric devices using graphene. The discussions for the experimental results are careful. I think that it is easy for readers to understand the article. Therefore, it is thought that this article is worth being placed in the Nanomaterials.
Reviewer 3 Report
The paper is interesting and well written. I think that the English quality should be improved.
The abstract does not summarizes completely what is done and described in this study. The process used should be better summarized in the abstract.
Line 26: concerns should be changed into properties.
Line 52: ", including metal catalysis..." This is not clear. What do the author mean?
In the experimental section the whole deposition process should be better described with thais of a drawing or a sketch.
Throughout the text there are several words that should be changes or corrected:
Line 78: components should be composition
Line 142: "Ni1Cu3 alloy has grown full .." The phrase should be rewritten in good English
Paragraph 3.3 This section should be briefly mentioned and summarized in the abstract
